# Likelihood of infectious diseases due to lack of exclusive breastfeeding among infants in Bangladesh

**Faruq Abdulla**[1]*, **Md. Moyazzem Hossain**[2,3], **Md. Karimuzzaman**[3], **Mohammad Ali**[4], **Azizur Rahman**[5]*

**1** Department of Applied Health and Nutrition, RTM Al-Kabir Technical University, Sylhet, Bangladesh, **2** School of Mathematics, Statistics & Physics, Newcastle University, Newcastle upon Tyne, United Kingdom, **3** Department of Statistics, Faculty of Mathematical and Physical Sciences, Jahangirnagar University, Savar, Dhaka, Bangladesh, **4** Centre for tropical medicine and global health, Nuffield Department of Medicine, University of Oxford, United Kingdom, **5** School of Computing, Mathematics and Engineering, Charles Sturt University, Wagga Wagga, NSW, Australia

* faruqiustat09mnil@gmail.com (FA); azrahman@csu.edu.au (AR)

**Data Availability Statement:** After registration, the data set is available via the following access link http://dhsprogram.com/data/available-datasets.cfm.

## Abstract

### Background

Bangladesh is a South Asian developing country trying to achieve the Sustainable Development Goals (SDG)-3 and the objective of the Rural Electrification Board (REB) regarding child mortality. Infectious diseases are leading causes of child mortality, and lack of exclusive breastfeeding (EBF) among infants aged 0–6 months increases child morbidity and mortality from various infectious diseases in developing countries. However, as per existing literature, no study has been conducted yet to determine the lack of EBF practice effect on child mortality in Bangladesh. With this backdrop, the authors intend to measure the likelihood of infectious diseases due to the lack of EBF of infants aged 0–6 months in Bangladesh.

### Materials and methods

This study used Bangladesh Demographic and Health Survey (BDHS) data over 1996–97 to 2017–18. The mothers of infants aged 0–6 months who were willingly participated in the BDHSs were considered to include in our analysis. Initially, there were 9,133 cases in the combined dataset. After filtering, there were 5,724 cases in the final dataset. We have considered diarrhea (D), acute respiratory infection (ARI) separately as well as the presence of either D or ARI or both and named as CoDARI as outcome variables. This study used both graphical and statistical techniques (Chi-square test, Wald test, and logistic regression) to analyze the data. The odds ratio (OR) and 95% confidence interval (CI) were used to quantify the likelihood of infectious diseases due to lack of EBF practice and its elasticity, respectively.

### Results

The EBF practice got a conspicuous increasing trend, but the prevalence of infectious diseases was declined from 0 to 3 months of age of infants, whereas an inverse scenario is

**Funding:** The authors received no specific funding for this work.

**Competing interests:** The authors have declared that no competing interests exist.

**Abbreviations:** ARI, Acute Respiratory Infection; BCG, Bacillus Calmette-Guérin; BDHS, Bangladesh Demographic and Health Survey; BF, Breastfeeding; CI, Confidence Interval; CoDARI, Combination of diarrhea or ARI (i.e. Diarrhea (D) or ARI or both); D, Diarrhea; EBF, Exclusive Breastfeeding; LRM, Logistic Regression Model; MICS, Multiple Indicator Cluster Survey; OR, Odds Ratio; PARI, Prevalence of ARI; PD, Prevalence of Diarrhea; PEBF, Prevalence of EBF; PCoDARI, Prevalence of the combination of diarrhea and ARI; REB, Rural Electrification Board; SDG, Sustainable Development Goals.

observed between 4–6 months. The significance of that inverse relationship was confirmed by p-value corresponding to the chi-square test and the Wald test of the adjusted regression coefficients after adjusting the associated factor's effect on infectious diseases. The adjusted ORs also concluded that the lack of EBF practice up to six months of age could enhance the risk of D, ARI, and CoDARI by 2.11 [95% CI: 1.56–2.85], 1.43 [95% CI: 1.28–1.60], and 1.48 [95% CI: 1.32–1.66] times higher, respectively.

## Conclusion

Findings of this study emphasize the importance of EBF up to six months of age of infants against diarrhea and ARI specific morbidity and mortality. Our results also agreed to the recommendation of the World Health Organization (WHO), United Nations International Children's Emergency Fund (UNICEF), American Academy of Pediatrics (AAP), American Academy of Family Physicians (AAFP), and National Nutrition Programme of Ethiopia (NNPE) that the EBF practice for the first six months of age could be a best, cost-effective, long-lasting natural preventive way to reduce the child morbidity and mortality due to infectious diseases in developing countries. Therefore, findings would help policymakers ensuring the achievement target of REB and SDG-3 associated with the health sector in Bangladesh.

## Introduction

Neonatal, infant, and child mortality are ongoing public health problems throughout the world. These are crucial indicators of a country's well-being and socio-economic development and are, therefore, used for monitoring and evaluating population and health programs and policies. However, deaths from infectious diseases significantly contribute to neonatal, infant, and child mortality rates. Infectious diseases are responsible for 7 out of 10 childhood deaths globally, where specifically, ARI is the leading cause for 30% of total childhood deaths, and diarrhea is the second leading cause of childhood deaths [1, 2]. Among the total, 95% of pneumonia cases occur in developing countries, and, therefore, newborns are more likely to die from infectious diseases from these countries.

Bangladesh is a South Asian developing country where a majority of people are poor. Over the last 25 years, the neonatal, infant, and child (under-5) mortality rates were declined, and, as per BDHS 2014, the neonatal, infant, and under-5 mortality rates were 30, 38, and 45 per 1000 live birth respectively, in Bangladesh [3]. However, Bangladesh is so far from attaining the objective of the Rural Electrification Board (REB) to lessen infant mortality by 5 per 1000 live birth [4, 5]. Moreover, substantial reduction is required in neonatal mortality to 12 per 1000 live births and under-5 mortality to as low as 25 per 1000 live births by the year 2030 to achieve the Sustainable Development Goal-3 (SDG-3) regarding child mortality [4, 6]. However, to avert the child mortality rate, the WHO, UNICEF, AAP, AAFP, and NNPE recommends timely commencement of breastfeeding within the first hour of birth and followed by breastfeeding exclusively for the first six months of age, and then continued breastfeeding along with other complementary foods up to 24 months of age in order to ensure the optimal growth, health and development [7–14]. There is evidence from many studies that EBF can play a significant role in boosting immunity and reducing the risk of morbidity and mortality of tremendous communicable and non-communicable diseases in the early and an older stage

if the duration of EBF is maintained properly [9, 15–17]. Furthermore, EBF up to six months of life of infants can avert the risk of diarrhea and ARI [18–20]. The infants who were not exclusively breastfeed had a higher likelihood of suffering from infectious diseases than those who received breast milk exclusively up to six months of age [21–23].

Due to non-exclusive or inappropriate or lack of EBF, about 1.24 million (96%) child deaths occur during the first six months of age, and the rate is higher in Asia and Africa [24, 25]. About 45% of neonatal infectious deaths, 30% diarrheal deaths, 18% acute respiratory deaths, and 10% disease burden of infants under five years of age are attributed to suboptimal breastfeeding in developing countries [26, 27]. Each year, more than 236,000 infant deaths occur for inadequate breastfeeding in China, India, Nigeria, Mexico, and Indonesia [28]. Therefore, the World Health Organization (WHO) recommends that all the countries should meet 90% EBF to reduce child death among under-5 children [29], and according to Sustainable Development Goals (SDGs) strategy, it is needed to rise 50% by 2025 [30, 31]. Many studies showed that about 1.5 million children's lives could be saved each year if they practiced by recommended EBF rates [32]. Moreover, some studies reveal that 13% - 15% of infant death among lives aged under-5 years could be reduced by increasing the EBF rate to an optimal level in low- and middle-income countries [29, 33].

Furthermore, Bangladesh is still now far away from the recommended percentage of the EBF. As per existing literature, till now, no study has been conducted yet to determine the effect of the lack of EBF practice on child mortality in Bangladesh. With this backdrop, the study was aimed to measure the likelihood of infectious diseases of infants aged 0–6 months due to lack of EBF practice using the Bangladesh Demographic and Health Survey (BDHS) data from 1996–97 to 2017–18. The findings of this study would help to make proper decisions against reducing childhood mortality and morbidity from infectious diseases through improving the prevalence of EBF practice.

## Materials and methods

### Study flow diagram

Initially, the comparative graphical analysis was performed to compare the patterns of EBF and infectious disease (D, ARI and CoDARI) over the survey years and infant's age to introduce their relationship. Subsequently, the chi-square test was carried out to determine the significance of the pre-determined relationship as well as to determine the significant associated factors other than EBF of infectious disease. Furthermore, this study was performed the simple logistic regression modeling to estimate the strength of the relationship and the likelihood of infectious diseases due to lack of EBF practice. Finally, the multiple logistic regression modeling was carried out to adjust the effect of other significant associated factors of infectious diseases. The study flow diagram is presented in Fig 1.

### Study data

The Bangladesh Demographic and Health Survey is the longest-running series of nationally representative household surveys in Bangladesh that started in 1993, collecting pooled population, health, and nutrition data. This study used BDHS data over 1996–97 to 2017–18 to complete the objectives.

### Case inclusion criteria

The mothers of infants aged 0–6 months who were willingly participated in the BDHSs were considered to include in our study.

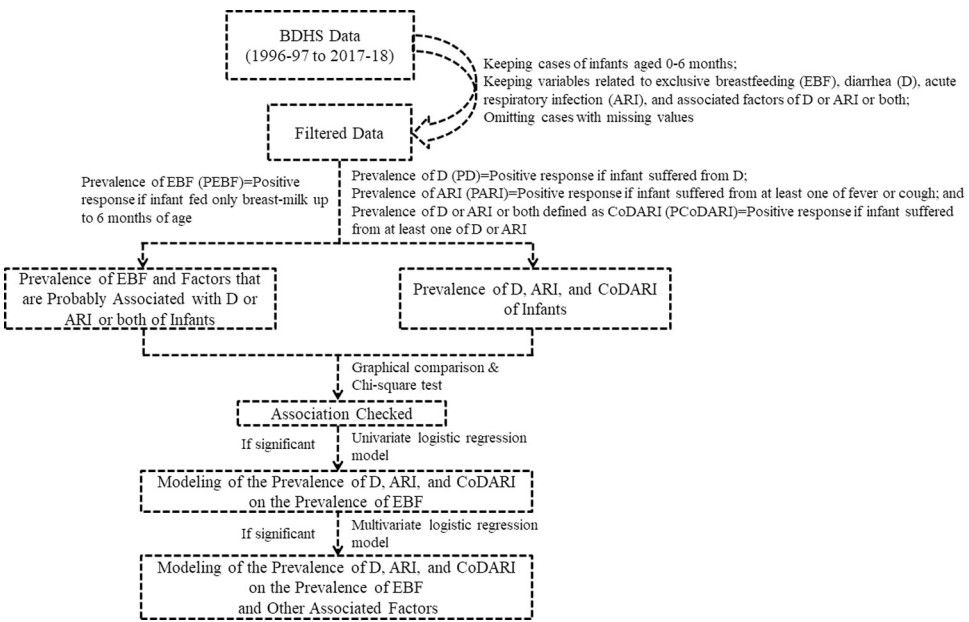

**Fig 1. Study flow diagram.**

## Outcome variable

Children are affected by several infectious diseases; however, we have considered diarrhea (D), ARI and the combination of both i.e., having either D or ARI or both as CoDARI (i.e., combination of D or ARI), therefore, infectious diseases indicate the presence of D, ARI, and CoDARI in this study. The interest of the study was to measure the likelihood of infectious diseases of infants aged 0–6 months due to lack of EBF practice; therefore, the prevalence of D, ARI, and CoDARI were considered as the outcome variables. All outcome variables considered in this study with their corresponding value labels are provided in the S1 Table.

## Predictor variable

The key predictor variable was the prevalence of EBF-computed using the related variables on feeding breast milk and complementary foods. Identifying the sole effect of EBF on infectious diseases needs to adjust the effect of other covariates, including socio-demographic, vaccination, and other related factors that may impact the risk of infectious diseases. The survey year was also considered as an associated factor to adjust the time effect. In addition, some covariates were re-coded before being used. However, the predictor variable and associated factors with their respective value labels are presented in the S1 Table.

## Prevalence of EBF and infectious diseases

The original survey of BDHS collected data on the variables related to current feeding status, complementary foods, diarrhea, and ARI (S1 Table). In order to perform the analysis, a new variable was defined for computing the prevalence of EBF and categorized as positive response identified by '1' if infant currently breastfed and given none of the complementary foods and categorized as negative response identified by '0' if currently breastfed and given at least one complementary food or currently not breastfed. Similarly, the prevalence of diarrhea was defined as a positive (assigned the value '1') or negative (assigned the value '0') response depending on whether the infant had diarrhea or not. Similarly, another new variable was

defined for the prevalence of ARI and categorized as a positive response (assigned by '1') if the infant suffered from fever and/or cough; otherwise, negative response (identified by '0'). Finally, a new variable was defined for the prevalence of CoDARI in infants, with a positive response denoted by a '1' indicating that the infant had at least one of diarrhea or ARI and a negative response denoted by a '0' indicating that the infant had none of the infectious diseases.

### Data processing

Firstly, the data sets were filtered with the inclusion criteria of an infant aged under six months and pre-selected plausible variables (see, S1 Table). After filtering the datasets, there were 9,133 remaining cases. Following that, cases with missing values were discarded, and the variables for the prevalence of EBF, D, ARI, and CoDARI were calculated using their respective definitions. After removing unnecessary variables, the data sets were combined by adding a variable for the survey year. There were 5,724 cases in the final dataset. The summary of the datasets is presented in the S2 Table.

### Statistical analysis

The Chi-square test was used to determine the significance of the bivariate association between variables. The simple and multiple modeling were done with the help of logistic regression model (LRM). A detailed history of the logistic regression is described by researchers [34, 35]. The entry method was employed in logistic regression modelling. The significance of the parameters was tested by the p-value of their respective Wald test. The OR and 95% CI were used to quantify the likelihood of infectious diseases due to lack of EBF practice and its elasticity, respectively. The data processing and analyses were done using IBM SPSS v25 and R-software.

## Results

Among 5724 infants, 1662 (29%) and 4062 (71%) were from urban and rural areas, respectively, along with the sex distribution as male 51.5% and female 48.5%. The percent distribution of infants by geographical divisions were found as Barisal (641, 11.2%), Chittagong (1207, 21.1%), Dhaka (1097, 19.2%), Khulna (674, 11.8%), Mymensing (132, 2.3%), Rajshahi (836, 14.6%), Rangpur (321, 5.6%), and Sylhet (816, 14.3%). Around one-third of the infant's parents had a secondary level of education, while around a quarter of the infant's parents had no formal education. However, more than 10% of the infant's parents had a higher educational background. The age distribution of the infants were 0–2 months (2427, 42.4%), 3–4 months (1644, 28.8%), and 5–6 months (1653, 28.9%) with mean and standard deviation were 3.07 and 1.93 months, respectively.

The overall breastfeeding trend seemed to be steady with nearly 100% during each survey period; whereas, a significant fluctuation was observed in EBF over the study period. The prevalence of EBF was approximately 40% in 1996–97 and 1999–00, with a slight decline at 2003–04. A sharp increase was observed after 2003–04, reached a peak of approximately 56% in 2011, followed by a declined trend in later years. Interestingly, the prevalence of infectious diseases have fluctuated inversely with EBF up to 2011, but after that, they show the same patterns (Fig 2(A)).

Comparing insight from each survey year revealed that EBF appeared to be more prevalent during the first three months of an infant's life, with an inverse scenario after three months of age. The results demonstrated that the prevalence of EBF (PEBF) increased up to three months (for most of the years); however, it decreased after the infant reached two or three months of

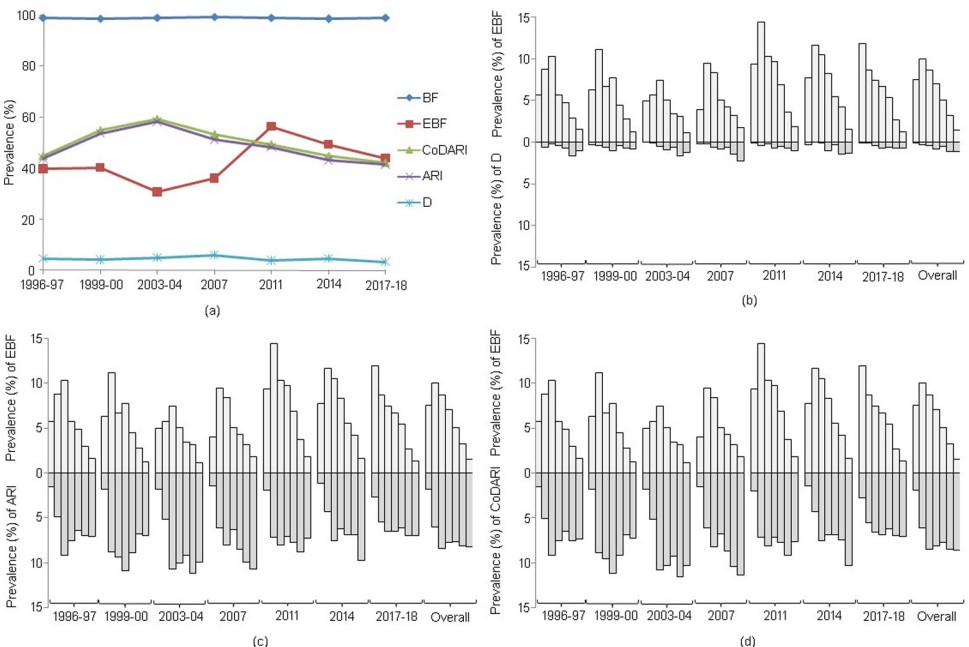

**Fig 2. Graphical comparison of the prevalence of D, ARI, and CoDARI with the prevalence of breastfeeding (BF) and EBF.** The figure (a) shows the trend of BF, EBF, D, ARI, and CoDARI over the survey years. The comparison of D, ARI, and CoDARI with EBF are shown in (b), (c), and (d), respectively according to child age in months. In figures (b)-(d), the histogram corresponding to each survey year represents the prevalence for 0 to 6 months of age from left to right.

age. Moreover, the sixth month of age had the lowest PEBF (1.13% to 1.83%) practice across all survey years (Fig 2B–2D). Intriguingly, the prevalence of diarrhea (PD) was the lowest (mean: 0.18%) at the infant's initial age (0 to 1 month) in each survey year. Additionally, the last three months (4 to 6 months) appeared critical as the PD increased as EBF declined. The last two months (5 to 6) of an infant's life were associated with the highest prevalence of diarrhea (mean: 1.14%) in each survey year (Fig 2(B)). Likewise, the lowest prevalence of ARI (PARI) (mean: 3.97%) and the prevalence of CoDARI (PCoDARI) (mean: 4.04%) were detected at the earliest ages, while the highest prevalence (PARI, mean: 8.22%; CoDARI, mean: 8.55%) was observed in infants aged 5 to 6 months (Fig 2C and 2D). In every survey year except the first two (1996–97 and 1999–00), the highest prevalence of ARI and prevalence of CoDARI were observed in the last two months (5 and 6) of infants with a low prevalence of EBF (Fig 2C and 2D).

The above-mentioned comparative graphical analysis of patterns of infectious diseases along with EBF provided a clear indication about a premise that there was a strong association between them. Therefore, the chi-square tests were conducted separately for each survey year and combinedly to validate that premise. The 0.1 or 10% significance level was used as the cut-off point for the p-value for statistical significance. Except for ARI in 2007 and diarrhea in 2011, the results of each survey year demonstrated a significant strong association between the practice of EBF and the prevalence of D, ARI, and CoDARI. Besides, considering all survey years combinedly, the results indicated that all of D, ARI, and CoDARI were significantly associated with EBF practice.

Furthermore, at a 10% level of significance the chi-square test confirmed that specific associated factors other than lack of EBF practice could significantly provoke the risk of D, ARI, and CoDARI. Also, the results indicated that BCG vaccination status significantly impacted

the likelihood of infectious diseases for most of the survey years. The risks of D, ARI, and CoDARI were significantly associated with the region, the respondent's education level, the source of drinking water, the type of floor material, the education and occupation of the respondent's partner, and the sex of child (Table 1). In addition, the type of toilet facility and type of resident significantly affected the likelihood of D and ARI as their corresponding p-values were less than the considered level of significance. Hence, the risk of infectious diseases could also be significantly changed over survey year/time.

Therefore, a logistic regression analysis was conducted to determine the sole effect of the lack of EBF practice on the risk of D, ARI, and CoDARI. Table 2 summarizes unadjusted and adjusted logistic regression results, where the parameter's significance was determined at a 10% level of significance. The significance test of the crude regression coefficient of the lack of EBF practice demonstrated that it had a significant effect on D, ARI, and CoDARI, except for ARI in 2007 and diarrhea in 2011 (Table 2). The crude coefficients of the lack of EBF practice on ARI and D in 2007 and 2011 were statistically near to significant. Despite adjusting for significant associated factors mentioned in Table 1, the adjusted coefficients of the lack of EBF practice were also highly significant in predicting the risk of D, ARI, and CoDARI, except for a few survey years (Table 2).

Finally, the ORs were calculated along with a 95% confidence interval to measure the effect of the lack of EBF practice on the likelihood of D, ARI, and CoDARI. The crude ORs for the lack of EBF practice on diarrhea ranged from 1.30 to 4.69, and the combined OR was 2.43. This means that infants who did not receive EBF until the age of six months had a 1.30 to 4.69 times greater risk of suffering from diarrhea than those who received it (Fig 3). Similarly, if infants were not exclusively breastfed for six months, they would have a 1.27 to 1.84 times greater likelihood of developing ARI and a 1.37 to 1.93 times greater chance of contracting an CoDARI (Fig 3). On the other hand, after adjusting for the effects of other significant associated factors, the adjusted ORs were not significantly different from the crude ORs. So, the adjusted ORs indicated that not practicing EBF until six months of age could significantly increased the risk of developing diarrheal diseases (about 1.27 to 3.92 times greater), ARI (about 1.13 to 1.66 times greater), and CoDARI (about 1.22 to 1.75 times greater) (Fig 3).

## Discussion

Lack of EBF practice during the first six months of life of infants enhances their risk of morbidity and mortality from infectious diseases. Therefore, the study was aimed to assess the protective capacity of EBF practice up to the age of six months against infectious diseases in Bangladesh. However, up to the first three months of the infant's life, the graphical comparisons revealed an increased prevalence of EBF, however, a decreased prevalence of D, ARI, and CoDARI, whereas an inverse scenario was observed for the later three months- the Chi-square and Wald tests justified the validity of their inverse relationship. A study examined the prevalence of EBF and infectious diseases in infants under three months of age using the Multiple Indicator Cluster Survey (MICS)-2003 data and showed that both diseases were significantly associated with the dearth of EBF practice [36]. Additionally, their study found that infants under three months who were exclusively breastfed had a lower risk of developing diarrhea and/or ARI. At 3–6 months of age, a longitudinal study discovered an inverse relationship between EBF and ARI with fever and gastrointestinal infection, concluding that EBF is protective against various ARI and gastrointestinal infections [37]. Several studies also found similar findings [38, 39]. Breast milk is the sole natural and primary source of optimum sustenance for newborn babies' physical and neurological growth and cognitive development [40]; it also boosts the child's immune system at an early age [41–43]. However, it may be challenging for

**Table 1. Significance of the association of infectious diseases with EBF and other associated factors.**

| Outcomes | Survey Year | | | | | | | | Associated factors |
|---|---|---|---|---|---|---|---|---|---|
| | 1996–97 | 1999–00 | 2003–04 | 2007 | 2011 | 2014 | 2017–18 | Overall | |
| D | 9.76 (0.002) | 5.74 (0.017) | 8.50 (0.004) | 3.93 (0.047) | 0.56 (0.454) | 4.11 (0.043) | 5.48 (0.019) | 36.65 (<0.001) | EBF (No, Yes) |
| ARI | 6.81 (0.009) | 15.87 (<0.001) | 13.98 (<0.001) | 2.08 (0.150) | 6.33 (0.012) | 17.16 (<0.001) | 19.96 (<0.001) | 87.66 (<0.001) | |
| CoDARI | 8.67 (0.003) | 17.13 (<0.001) | 17.96 (<0.001) | 3.57 (0.059) | 7.05 (0.008) | 18.72 (<0.001) | 21.41 (<0.001) | 101.25 (<0.001) | |
| D | 0.99 (0.320) | 5.96 (0.015) | 0.00 (0.961) | 0.45 (0.501) | 0.83 (0.362) | 0.08 (0.772) | 0.00 (0.960) | 0.81 (0.368) | Type of Resident (Urban, Rural) |
| ARI | 0.58 (0.448) | 0.15 (0.699) | 0.50 (0.479) | 0.59 (0.443) | 0.20 (0.658) | 1.28 (0.257) | 0.10 (0.755) | 1.86 (0.173) | |
| CoDARI | 0.16 (0.688) | 0.00 (0.964) | 0.82 (0.364) | 0.26 (0.608) | 0.17 (0.684) | 1.19 (0.276) | 0.07 (0.798) | 0.78 (0.377) | |
| D | 6.76 (0.239) | 4.55 (0.473) | 10.78 (0.056) | 1.28 (0.937) | 7.37 (0.288) | 3.70 (0.717) | 7.78 (0.352) | 5.96 (0.544) | Region (Barisal, Chittagong, Dhaka, Khulna, Mymensingh, Rajshahi, Rangpur, Sylhet) |
| ARI | 14.92 (0.011) | 5.31 (0.379) | 7.87 (0.163) | 5.63 (0.344) | 14.37 (0.026) | 6.74 (0.346) | 8.21 (0.314) | 14.78 (0.039) | |
| CoDARI | 16.91 (0.005) | 5.46 (0.362) | 5.73 (0.334) | 4.94 (0.423) | 14.60 (0.024) | 8.30 (0.217) | 7.93 (0.339) | 15.03 (0.036) | |
| D | 3.74 (0.291) | 11.97 (0.007) | 2.75 (0.432) | 6.46 (0.091) | 2.32 (0.510) | 0.58 (0.901) | 1.55 (0.670) | 4.23 (0.238) | Respondent's Education Level (No education, Primary, Secondary, Higher) |
| ARI | 7.02 (0.071) | 1.42 (0.701) | 3.77 (0.287) | 5.32 (0.150) | 4.21 (0.239) | 3.12 (0.373) | 6.85 (0.077) | 20.24 (<0.001) | |
| CoDARI | 7.65 (0.054) | 2.34 (0.505) | 3.39 (0.336) | 5.86 (0.118) | 3.66 (0.301) | 2.59 (0.459) | 6.17 (0.104) | 17.13 (0.001) | |
| D | 0.25 (0.884) | 0.34 (0.846) | 2.69 (0.261) | 1.28 (0.527) | 0.22 (0.896) | 0.10 (0.949) | 5.27 (0.072) | 2.02 (0.364) | Source of Drinking Water (Piped, Tubewell, River/ Pond/Surface/Rain/etc) |
| ARI | 0.60 (0.742) | 0.10 (0.949) | 2.49 (0.288) | 6.58 (0.037) | 0.50 (0.781) | 1.78 (0.412) | 4.89 (0.087) | 18.43 (<0.001) | |
| CoDARI | 1.14 (0.565) | 0.26 (0.877) | 2.05 (0.358) | 4.72 (0.094) | 0.25 (0.881) | 1.41 (0.493) | 5.94 (0.051) | 16.88 (<0.001) | |
| D | 0.28 (0.965) | 4.54 (0.209) | 2.54 (0.468) | 0.77 (0.857) | 2.15 (0.542) | 4.04 (0.258) | 3.72 (0.294) | 1.22 (0.748) | Type of Toilet Facility (Modern, Pit latrine, Others type latrine, No facility) |
| ARI | 2.03 (0.565) | 3.57 (0.312) | 3.82 (0.281) | 4.77 (0.189) | 1.24 (0.745) | 1.30 (0.730) | 2.60 (0.458) | 6.76 (0.080) | |
| CoDARI | 1.97 (0.579) | 3.61 (0.307) | 4.19 (0.242) | 3.31 (0.346) | 0.59 (0.899) | 0.45 (0.930) | 3.70 (0.296) | 4.09 (0.252) | |
| D | 2.81 (0.245) | 0.70 (0.402) | 0.94 (0.333) | 1.13 (0.570) | 0.23 (0.891) | 0.54 (0.762) | 4.96 (0.084) | 2.60 (0.273) | Floor Material (Katcha, Pacca, Others) |
| ARI | 0.80 (0.670) | 5.18 (0.023) | 1.41 (0.236) | 6.31 (0.043) | 6.07 (0.048) | 3.31 (0.191) | 3.73 (0.155) | 38.61 (<0.001) | |
| CoDARI | 0.92 (0.632) | 4.00 (0.045) | 1.34 (0.246) | 5.49 (0.064) | 5.13 (0.077) | 3.58 (0.167) | 5.06 (0.080) | 36.71 (<0.001) | |
| D | 3.51 (0.319) | 9.46 (0.024) | 5.92 (0.116) | 4.52 (0.211) | 1.12 (0.772) | 6.87 (0.076) | 1.10 (0.776) | 4.25 (0.235) | Partner's Education (No education, Primary, Secondary, Higher) |
| ARI | 4.15 (0.246) | 4.90 (0.179) | 12.59 (0.006) | 4.11 (0.250) | 2.04 (0.564) | 3.82 (0.282) | 4.05 (0.256) | 30.75 (<0.001) | |
| CoDARI | 4.86 (0.183) | 3.71 (0.295) | 12.79 (0.005) | 5.90 (0.116) | 2.64 (0.450) | 3.22 (0.360) | 4.25 (0.236) | 31.07 (<0.001) | |

(*Continued*)

**Table 1.** (Continued)

| Outcomes | Survey Year | | | | | | | | Associated factors |
|---|---|---|---|---|---|---|---|---|---|
| | 1996–97 | 1999–00 | 2003–04 | 2007 | 2011 | 2014 | 2017–18 | Overall | |
| D | 14.35 (0.045) | 29.97 (<0.001) | 5.74 (0.570) | 10.07 (0.185) | 8.14 (0.320) | 6.66 (0.465) | 8.80 (0.268) | 11.06 (0.136) | Partner's Occupation (Agriculture, Non-agriculture, Unskilled, Skilled, Professional, Big business, Small business, Others) |
| ARI | 9.26 (0.235) | 13.71 (0.057) | 6.86 (0.443) | 5.06 (0.653) | 10.46 (0.164) | 12.38 (0.089) | 7.06 (0.423) | 31.72 (<0.001) | |
| CoDARI | 10.56 (0.159) | 12.49 (0.085) | 6.01 (0.539) | 3.65 (0.819) | 10.19 (0.178) | 14.32 (0.046) | 7.40 (0.388) | 30.92 (<0.001) | |
| D | 0.09 (0.771) | 1.81 (0.179) | 0.47 (0.495) | 0.67 (0.412) | 3.39 (0.065) | 0.30 (0.583) | 0.00 (0.952) | 0.79 (0.374) | Sex of Child (Male, Female) |
| ARI | 0.32 (0.574) | 12.03 (0.001) | 1.06 (0.303) | 8.05 (0.005) | 5.20 (0.023) | 7.31 (0.007) | 5.84 (0.016) | 23.21 (<0.001) | |
| CoDARI | 0.30 (0.583) | 11.08 (0.001) | 1.08 (0.299) | 6.91 (0.009) | 4.79 (0.029) | 5.63 (0.018) | 5.10 (0.024) | 19.93 (<0.001) | |
| D | 7.07 (0.008) | 9.76 (0.002) | 5.46 (0.019) | 7.41 (0.006) | 0.41 (0.523) | 2.07 (0.150) | 4.82 (0.028) | 30.85 (<0.001) | Received Bacillus Calmette-Guérin (BCG) vaccine (Not received BCG, Received BCG) |
| ARI | 22.90 (<0.001) | 19.79 (<0.001) | 17.94 (<0.001) | 5.68 (0.017) | 17.97 (<0.001) | 18.91 (<0.001) | 54.72 (<0.001) | 127.88 (<0.001) | |
| CoDARI | 26.51 (<0.001) | 24.16 (<0.001) | 21.43 (<0.001) | 6.50 (0.011) | 18.96 (<0.001) | 19.12 (<0.001) | 57.92 (<0.001) | 143.54 (<0.001) | |
| D | | | | | | | | 9.25 (0.160) | Survey Year (1996–97, 1999–00, 2003–04, 2007, 2011, 2014, 2017–18) |
| ARI | | | | | | | | 77.82 (<0.001) | |
| CoDARI | | | | | | | | 79.18 (<0.001) | |

women to stick to EBF for six months, especially in low and lower-middle income countries like Bangladesh, where maternal malnutrition is frequent, leading to reduce breast milk production [44–48]. In addition, a lack of information about the benefits of EBF practice, insufficient workplace supports for mothers, and insufficient healthcare system support all contribute to mothers discontinuing the EBF practice before the recommended six-month period [49]. Previous studies pointed out that EBF practice for up to 6 months could have prevented the occurrences of diarrhea and ARI [49, 50]. The reason for working behind may be that the body gradually builds prevention ability with the help of EBF and natural immunity [49]. Also, without EBF, children are given foods and fluids that are potentially contaminated and/or difficult to digest [21, 51].

As this study was intended to find the sole effect of EBF on infectious diseases, the researcher was concerned to adjust the influences of other co-factors on infectious diseases. The existing literature suggested that several socio-demographic, geospatial, environmental factors, and vaccination status are controlling factors of infectious diseases. The findings revealed that the selected co-factors have a significant contribution to the prevalence of infectious diseases of infants, which is consistent with previous study results [46, 49, 51–54]. The severity of most infectious diseases in males can be explained by the impact of sex hormones on the T-helper 1/T-helper 2 cytokines, therefore, the female counterparts have higher morbidity and mortality because of greater immunopathology and/or autoimmunity [55]. Vaccination with BCG can develop cross-protective immunity by inducing an improved innate immune response for trained immunity against different microorganisms other than Mycobacterium tuberculosis [56] and consumption of vitamin A and different vaccination reduce childhood illness [57, 58], however, the life-saving vaccination coverage has been disrupted by

**Table 2. Results of logistic regression models (crude and adjusted) of infectious diseases on EBF.**

| Outcome Variables | | Crude Estimated β of EBF | SE | p-value | Adjusted Estimated β of EBF | SE | p-value | Survey Year |
|---|---|---|---|---|---|---|---|---|
| Prevalence (%) | D | 1.546 | 0.542 | 0.004 | 1.367 | 0.551 | 0.013 | 1996–97 |
| | ARI | 0.414 | 0.159 | 0.009 | 0.265 | 0.169 | 0.119 | |
| | CoDARI | 0.466 | 0.159 | 0.003 | 0.311 | 0.170 | 0.067 | |
| | D | 0.993 | 0.430 | 0.021 | 0.774 | 0.450 | 0.086 | 1999–00 |
| | ARI | 0.569 | 0.143 | <0.001 | 0.509 | 0.151 | 0.001 | |
| | CoDARI | 0.592 | 0.144 | <0.001 | 0.519 | 0.152 | 0.001 | |
| | D | 1.436 | 0.533 | 0.007 | 1.332 | 0.544 | 0.014 | 2003–04 |
| | ARI | 0.578 | 0.155 | <0.001 | 0.490 | 0.163 | 0.003 | |
| | CoDARI | 0.657 | 0.156 | <0.001 | 0.561 | 0.163 | 0.001 | |
| | D | 0.788 | 0.407 | 0.053 | 0.511 | 0.419 | 0.222 | 2007 |
| | ARI | 0.239 | 0.166 | 0.150 | 0.121 | 0.177 | 0.494 | |
| | CoDARI | 0.313 | 0.166 | 0.059 | 0.201 | 0.176 | 0.254 | |
| | D | 0.261 | 0.350 | 0.455 | 0.239 | 0.350 | 0.495 | 2011 |
| | ARI | 0.334 | 0.133 | 0.012 | 0.198 | 0.142 | 0.164 | |
| | CoDARI | 0.352 | 0.133 | 0.008 | 0.216 | 0.142 | 0.128 | |
| | D | 0.743 | 0.374 | 0.047 | 0.724 | 0.376 | 0.054 | 2014 |
| | ARI | 0.611 | 0.148 | <0.001 | 0.474 | 0.157 | 0.003 | |
| | CoDARI | 0.635 | 0.147 | <0.001 | 0.499 | 0.156 | 0.001 | |
| | D | 0.886 | 0.390 | 0.023 | 0.677 | 0.400 | 0.091 | 2017–18 |
| | ARI | 0.560 | 0.126 | <0.001 | 0.370 | 0.133 | 0.005 | |
| | CoDARI | 0.578 | 0.126 | <0.001 | 0.383 | 0.132 | 0.004 | |
| | D | 0.887 | 0.151 | <0.001 | 0.747 | 0.154 | <0.001 | Overall |
| | ARI | 0.504 | 0.054 | <0.001 | 0.357 | 0.058 | <0.001 | |
| | CoDARI | 0.541 | 0.054 | <0.001 | 0.390 | 0.058 | <0.001 | |

Note: D: diarrhea, ARI: acute respiratory infections, CoDARI: Either D or ARI or both, SE: standard error, EBF: exclusive breastfeeding, 10% level of significance was considered.

COVID-19 in low-and middle-income countries [59]. Children from families who have no access to safe drinking water, modern sanitation facilities, and living room materials have a greater likelihood of suffering from the infectious disease since their mother acts as a bearer of different microorganisms that may be imported into their body [60–62]. The parent's knowledge, attitudes and practice (KAP) in baby care and EBF are linked with their education level [63]. The reason behind the significance of the survey year may be that different promotional activities have been taken over time, such as the World Breastfeeding Week celebration and extensive mass media programmes, helped to expand EBF practice significantly as well as enhanced knowledge and attitudes towards EBF among Bangladeshi women over time [44, 64].

The crude ORs exhibited that the infants who did not receive EBF up to six months of age had a higher risk of suffering from D, ARI, and CoDARI, respectively, than those infants who received EBF up to six months of age. The adjusted ORs also showed that infants who did not receive EBF until six months of age had 2.11 [95% CI: 1.56–2.85], 1.43 [95% CI: 1.28–1.60], and 1.48 [95% CI: 1.32–1.66] times greater likelihood of suffering from D, ARI, and CoDARI, respectively than infants who received EBF until six months of age. The findings corroborate those of several previous studies [21–25]. In comparison to EBF in the earliest age of infants, partial or no breastfeeding was associated with the increment in the risk of infant deaths from all causes, ARI, and diarrhea, respectively, in Dhaka slums [21]. Another research showed that

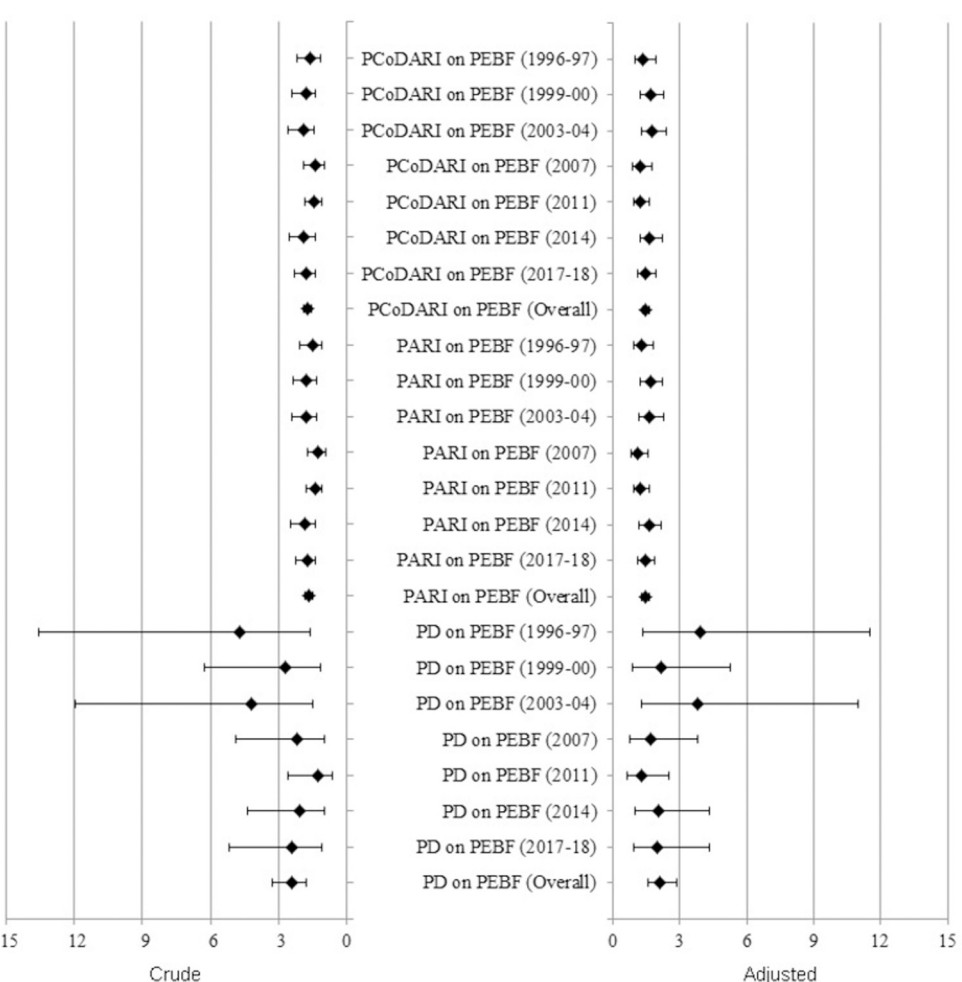

**Fig 3. Forest plot of crude and adjusted odds ratios and their respective 95% confidence intervals where PD, PARI, PCoDARI, and PEBF stands for prevalence of diarrhea, prevalence of ARI, prevalence of CoDARI, and prevalence of EBF, respectively.**

non-breastfed infants were associated with an increase in diarrhea incidence and mortality of 165% and 952% in infants aged 0–5 months, respectively, compared to EBF practice [22, 23]. Infants who were breastfed for the first year of life had a 30% lower risk of diarrhea, while infants primarily fed formula food had an 80% elevated risk of diarrhea [18, 19]. Infants who were breastfed exclusively for at least four months had a 72% lower risk of respiratory infection and a 74% lower risk of syncytial virus bronchiolitis [20]. The infants who exclusively breastfed for at least three months and six months had 50% and 63% lower risk of acute otitis media, respectively, and the risk of asthma, dermatitis, and eczema was reduced by approximately one-third of the infants exclusively breastfed for at least three months [20].

## Conclusion

In Bangladesh, approximately half of all children were breastfed exclusively for the first six months of their lives. Before 2011, the prevalence of D, ARI, and CoDARI were inversely connected with EBF, but after 2011, they were changed in the same direction. The findings of this study ensured the protective ability of EBF for the first six months of age of infants against

diarrhea and ARI-specific morbidity and mortality. Therefore, according to the outcomes of this study, the EBF practice up to the age of six months for newborns could be the best, cost-effective, and long-lasting natural child survival intervention in Bangladesh. This study was also estimated the risk of D, ARI, and CoDARI for infants aged 0–6 months due to lack of EBF practice up to their first six months of life in Bangladesh. The authors strictly recommend that the prevalence of EBF practice should be increased for attaining SDG-3. Therefore, to increase the prevalence of EBF practice, the authors advocate creating a favorable working environment for mothers. In addition, adopting appropriate labor laws relating to maternity care in public and private sectors in Bangladesh may significantly increase EBF practice. Finally, to broaden understanding, many EBF practice awareness activities at the individual and community levels are necessary to increase EBF practice. The authors believe that the findings of this paper will assist policymakers in accelerating the achievement of the SDG-3 and the REB's health sector objectives in Bangladesh.

## Supporting information

**S1 Table. List of variables with their respective definition and value labels.**
(DOCX)

**S2 Table. Summary of the datasets.**
(DOCX)

## Acknowledgments

The authors are grateful to ICF International, Rockville, Maryland, USA, for providing the Bangladesh DHS data sets for this analysis. We are also grateful to the well-wishers and their peers to motivate us for doing this research. Last but not least, the authors would like to sincerely thank the three reviewers, the Editor, and Academic Editor, for their valuable comments and suggestions, which have been used to improve the quality of the manuscript.

## Author Contributions

**Conceptualization:** Faruq Abdulla, Md. Moyazzem Hossain, Md. Karimuzzaman, Mohammad Ali, Azizur Rahman.

**Data curation:** Faruq Abdulla, Md. Moyazzem Hossain.

**Formal analysis:** Faruq Abdulla.

**Methodology:** Faruq Abdulla, Md. Moyazzem Hossain, Md. Karimuzzaman, Azizur Rahman.

**Supervision:** Azizur Rahman.

**Validation:** Azizur Rahman.

**Visualization:** Faruq Abdulla, Azizur Rahman.

**Writing – original draft:** Faruq Abdulla, Md. Moyazzem Hossain, Md. Karimuzzaman.

**Writing – review & editing:** Faruq Abdulla, Md. Moyazzem Hossain, Mohammad Ali, Azizur Rahman.

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
