## [Decision Letter · Decision Letter 0]

12 Aug 2021

PONE-D-21-16489

Likelihood of Infectious Diseases (Diarrhea/ARI) Due to Lack of Exclusive Breastfeeding of Infants (0-6 Months) in Bangladesh

PLOS ONE

Dear Dr. Rahman

Thank you for submitting your manuscript to PLOS ONE. After careful consideration, we feel that it has merit but does not fully meet PLOS ONE’s publication criteria as it currently stands. Therefore, we invite you to submit a revised version of the manuscript that addresses the points raised during the review process.

We look forward to receiving your revised manuscript.

Kind regards,

John M. Humphrey

Academic Editor

PLOS ONE

1. Please ensure that your manuscript meets PLOS ONE's style requirements, including those for file naming. The PLOS ONE style templates can be found at https://journals.plos.org/plosone/s/file?id=wjVg/PLOSOne_formatting_sample_main_body.pdf and https://journals.plos.org/plosone/s/file?id=ba62/PLOSOne_formatting_sample_title_authors_affiliations.pdf.

A clean copy of the edited manuscript (uploaded as the new *manuscript* file).

Additional Editor Comments (if provided):

Dear Dr. Rahman,

Please find below the reviewers' comments for your manuscript. I believe that these comments can be addressed and, in so doing, will improve the quality of the manuscript. Although the comments are extensive in both number and scope, in sum I believe that they still qualify as 'minor revision'. I look forward to reviewing a revised version of your manuscript with point-by-point response to the reviewers' comments.

Regards,

John Humphrey, MD, MS

Reviewers' comments:

Reviewer's Responses to Questions

**Comments to the Author**

1. Is the manuscript technically sound, and do the data support the conclusions?

Reviewer #1: Yes

Reviewer #2: Yes

Reviewer #3: Yes

2. Has the statistical analysis been performed appropriately and rigorously? 

Reviewer #1: Yes

Reviewer #2: Yes

Reviewer #3: Yes

3. Have the authors made all data underlying the findings in their manuscript fully available?

Reviewer #1: Yes

Reviewer #2: No

Reviewer #3: Yes

4. Is the manuscript presented in an intelligible fashion and written in standard English?

Reviewer #1: No

Reviewer #2: Yes

Reviewer #3: Yes

5. Review Comments to the Author

Reviewer #1: Summary: Authors have a seemingly well-conducted data analysis of the BDHS. The article is an appropriate one to discuss the role of EBF in the prevention of infectious diseases. The objective is focused and clearly mentioned.

The whole manuscript needs to be much more concise throughout. In the overall manuscript, it seems very confusing as Diarrhea, and ARI are not infectious diseases but Infectious diseases (ARI/Diarrhea) are different from the earlier two.

There are however some critical methodological and presentation considerations that might improve the manuscript greatly:

The title might be more precise like, “Likelihood of infectious diseases due to lack of exclusive breastfeeding among infants in Bangladesh”.

Abstract:

1. The meaning of the 1st line of the Background section is not clear.

2. Line 27: define SDG acronym at first use

3. Line 29: EBF, ARI define acronym at first use

4. In the result section: only adjusted ORs can be mentioned, authors may omit the crude values. It makes the result section clumsy.

5. Conclusion: Line 54 can be like, “Findings of this study emphasize the importance of EBF up to six months of age to prevent diarrhea and ARI………… “ and this complex line should be broken down into two simple sentences for better understanding of the reader.

6. Define the acronyms: WHO, UNICEF, AAP, AAFP, and NNPE.

Introduction:

1. Line 80: REB should be elaborated.

2. Line 96: can be like “The infants were not exclusively breastfed had a higher likelihood……….

3. “Moreover” – the term is used several times like, in line 109, 114…

4. Line 119-129: No idea why these pieces of information are in the Introduction section? Repetitive of the abstract.

Materials and Method:

1. Conceptual framework: can be replaced by a study flow diagram and “A conceptual framework illustrates the whole sequential procedure of a study.” This line can be removed.

2. Line 170, 171: currently fed breast-milk….. should be “currently breastfed…..”.

3. Statistical analysis: Recommend detailing the specific analyses

a. Line 199- 209: Logistic Regression Model section is not required here, either author may add a reference.

b. during modelling what method was followed in logistic regression analysis (entry, stepwise, etc), not clear?

4. It seems like when authors use infectious disease (Diarrhea/ ARI), but only “diarrhea” and only “ARI” are not infectious diseases!! The author needs to rename the variable “infectious disease (Diarrhea/ ARI)”.

Result: Overall the result section is not written in a standard manner, which is not up to the mark for a prestigious journal like PLOS One.

1. Figure title should be revised. What does it mean by “D, ID, (D/ARI)” should be mentioned.

2. Line 232, 235, 236: better to mention exact figure for prevalence like lowest prevalence (), highest prevalence ()…., it is difficult for the author to find out the prevalence from the table/ figure.

3. Line 242- 253: why a different p-value was considered, is not clearly mentioned.

4. Table 1 only presents p values which is a bit misleading

5. Table 1. Need to mention the comparison group (categories) among the independent variables. Like: type of residence- urban/ rural etc. Better to replace independent variables with “associated factors” and dependent variables with “outcomes”

6. Table 1: format should be changed, as it is difficult to understand the p values from 2 rows.

7. Line 305: what are the other significant factors which were adjusted?

8. Table 2:

a. needs to mention what is D, ID, ARI, coefficient beta, SE below the table.

b. instead of mentioning p value=0.000, better to use like <0.001.

c. better to mention the significance level.

9. Line: 325: omit “diseases”

Discussion: Use the discussion to detail how their findings add to the literature. The author just mentioned the similarities of their findings with other literature, but the reason behind those could be highlighted.

1. Need to elaborate MICS-2003.

2. There is no paragraph found for table 1 in the discussion section. Then why authors look for the significance values of the associated factors is unclear.

3. Line: 365-366, 369- 370: better to remove the ORs and 95% CIs from the discussion section.

4. Line 375: “not breastfeeding” should be replaced by “non-breastfed infants” are associated…..

5. Line 376- 377: omit the RR values.

6. It should not be recommended to use too many values in the discussion section.

Conclusion:

1. There is a repetition of some lines from the abstract and Introduction.

2. Need to add some lines as a recommendation.

References:

Ref 28: needs to be edited

Others:

English and grammar in the manuscript are relatively poor which obscures the readers' understanding throughout much of the work.

Reviewer #2: Thank you for the opportunity to review this manuscript. It is well written but authors have to work on the following

1. Avoid Abbreviation in the title

2. Any abbreviation has to be written in long form in the first time used

3. The abstract is unnecessarily long, some findings like the chi-square and crude odds ration can be reported in the results section in the main document.

4. In the material and method section a brief description of the conceptual framework is needed before authors refer the reader to the figure.

5. Table 1 needs to be presented in a more simplified way.

6. In table two present the odds ratio and confidence intervals

7. The discussion has a lot of repetition of the results, interpretation and discussion of results needs to be strengthen

Reviewer #3: Likelihood of Infectious Diseases (Diarrhea/ARI) Due to Lack of Exclusive Breastfeeding of Infants (0-6 months) in Bangladesh

This manuscript reports the findings from the quantitative analytical cross-sectional designed study which aimed to measure the likelihood of infectious diseases (diarrhea/ARI) due to lack of Exclusive Breast Feeding (EBF) of infants aged 0-6 months in Bangladesh. The need of this study is demonstrated by the slow reduction of neonatal mortality rate to achieve the SDG-3 and the evidence that most neonatal and infants infectious disease burden are attributed to suboptimal breastfeeding in developing countries.

This topic is of public health concern in developing countries. There are limited empirical studies in the region so this study has the potential to fill that gap. The strength of this study is the use of large data and the clear description of the method used which may allow the replication of the study.

6. PLOS authors have the option to publish the peer review history of their article (what does this mean?). If published, this will include your full peer review and any attached files.

Reviewer #1: No

Reviewer #2: **Yes: **Fabiola Vincent Moshi

Reviewer #3: **Yes: **Saada Ali Seif

---

## [Author Response · Author response to Decision Letter 0]

7 Sep 2021

Please refer to "Response to Reviewers.docx" file submitted with the revised version.

---

## [Decision Letter · Decision Letter 1]

31 Jan 2022

Likelihood of Infectious Diseases Due to Lack of Exclusive Breastfeeding among Infants in Bangladesh

PONE-D-21-16489R1

Dear Dr. Rahman,

We’re pleased to inform you that your manuscript has been judged scientifically suitable for publication and will be formally accepted for publication once it meets all outstanding technical requirements.

Kind regards,

Ricardo Q. Gurgel, PhD

Academic Editor

PLOS ONE

Additional Editor Comments (optional):

Reviewers' comments:

Reviewer's Responses to Questions

**Comments to the Author**

1. If the authors have adequately addressed your comments raised in a previous round of review and you feel that this manuscript is now acceptable for publication, you may indicate that here to bypass the “Comments to the Author” section, enter your conflict of interest statement in the “Confidential to Editor” section, and submit your "Accept" recommendation.

Reviewer #1: All comments have been addressed

Reviewer #3: All comments have been addressed

2. Is the manuscript technically sound, and do the data support the conclusions?

Reviewer #1: Yes

Reviewer #3: Yes

3. Has the statistical analysis been performed appropriately and rigorously? 

Reviewer #1: Yes

Reviewer #3: Yes

4. Have the authors made all data underlying the findings in their manuscript fully available?

Reviewer #1: Yes

Reviewer #3: Yes

5. Is the manuscript presented in an intelligible fashion and written in standard English?

Reviewer #1: Yes

Reviewer #3: No

6. Review Comments to the Author

Reviewer #1: After revision, the manuscript is written in well manner. Accept after major revision, there is no further comments.

Reviewer #3: I congratulate the authors for addressing all the comments raised. though need to work a little bit on Language issues. E.g. in line 37 and 119 and many other places used present tense instead of past tense

7. PLOS authors have the option to publish the peer review history of their article (what does this mean?). If published, this will include your full peer review and any attached files.

Reviewer #1: No

Reviewer #3: **Yes: **Saada Ali Seif

---

## [Editor Report · Acceptance letter]

7 Feb 2022

PONE-D-21-16489R1 

Likelihood of Infectious Diseases Due to Lack of Exclusive Breastfeeding among Infants in Bangladesh 

Dear Dr. Rahman:

I'm pleased to inform you that your manuscript has been deemed suitable for publication in PLOS ONE. Congratulations! Your manuscript is now with our production department. 

Kind regards, 

on behalf of

Professor Ricardo Q. Gurgel 

Academic Editor

PLOS ONE